# Metabolic and inflammatory axes of insulin resistance in normoglycemic adults with high HIV burden

Situmbeko Liweleya[1,2], Brian Halubanza[3], Lweendo Muchaili[1,2],
Bislom C. Mweene[1,2], Lukundo Siame[1,2], Propheria C. Lwiindi[1,2], Benson M. Hamooya[1],
Joreen P. Povia[4], Freeman M. Chabala[5], Musalula Sinkala[6,7,8], John R. Koethe[9],
Annet Kirabo[10,11,12,13], Sepiso K. Masenga[1,2,13,14] *

**1** HAND Research Group, School of Medicine and Health Sciences, Mulungushi University, Livingstone Campus, Livingstone, Zambia, **2** Department of Cardiovascular Science and Metabolic Diseases, Livingstone Center for Prevention and Translational Science, Livingstone, Zambia, **3** Department of Computer Science and Information Technology, School of Engineering and Technology, Mulungushi University, Kabwe, Zambia, **4** Department of Health Economics, Livingstone Center for Prevention and Translational Sciences, Livingstone, Zambia, **5** The Institute of Basic and Biomedical Sciences, Levy Mwanawasa Medical University, Lusaka, Zambia, **6** Division of Computational Biology, Department of Integrative Biomedical Sciences, Faculty of Health Sciences, University of Cape Town, Cape Town, South Africa, **7** Institute of Infectious Diseases and Molecular Medicine, Faculty of Health Sciences, University of Cape Town, Cape Town, South Africa, **8** Department of Biomedical Sciences, School of Health Sciences, University of Zambia, Lusaka, Zambia, **9** Division of Infectious Diseases, Vanderbilt University Medical Center, Nashville, Tennessee, United States of America, **10** Department of Medicine, Vanderbilt University Medical Center, Division of Genetics and Clinical Pharmacology, Nashville, Tennessee, United States of America, **11** Vanderbilt Center for Immunobiology, Nashville, Tennessee, United States of America, **12** Vanderbilt Institute for Infection, Immunology and Inflammation, Nashville, Tennessee, United States of America, **13** Vanderbilt Institute for Global Health, Nashville, Tennessee, United States of America, **14** Department of Molecular Physiology and Biophysics, Vanderbilt University, Nashville, Tennessee, United States of America

* sepisomasenga@gmail.com

## Abstract

Insulin resistance (IR) precedes type 2 diabetes and drives cardiovascular risk, but its early features in normoglycemic individuals remain unclear. The main goal of the study was to characterize multifactorial determinants of IR in normoglycemic adults. We conducted a retrospective cross-sectional analysis of a census of 100 normoglycemic adults (median age 44 years, interquartile range (IQR): 35–49), 70% (70/100) of whom were living with HIV and 65% (65/100) were obese (body mass index (BMI) ≥30 kg/m²). Fasting blood samples were profiled for amino acids, adipokines, and inflammatory markers using mass spectrometry and immunoassays. IR was assessed as a continuous variable using the Homeostatic Model Assessment of IR version 2 (HOMA2-IR). Multivariable linear regression identified independent factors associated with IR, stratified by obesity and adjusted for age and sex. In the overall cohort, the prevalence of IR was 36%. The adjusted analysis revealed leptin (β = 0.041, p = 0.009), triglycerides (β = 0.007, p = 0.006), glutamic acid (β = 0.025, p = 0.0001) and tyrosine (β = 0.029, p = 0.0004) were independently associated with

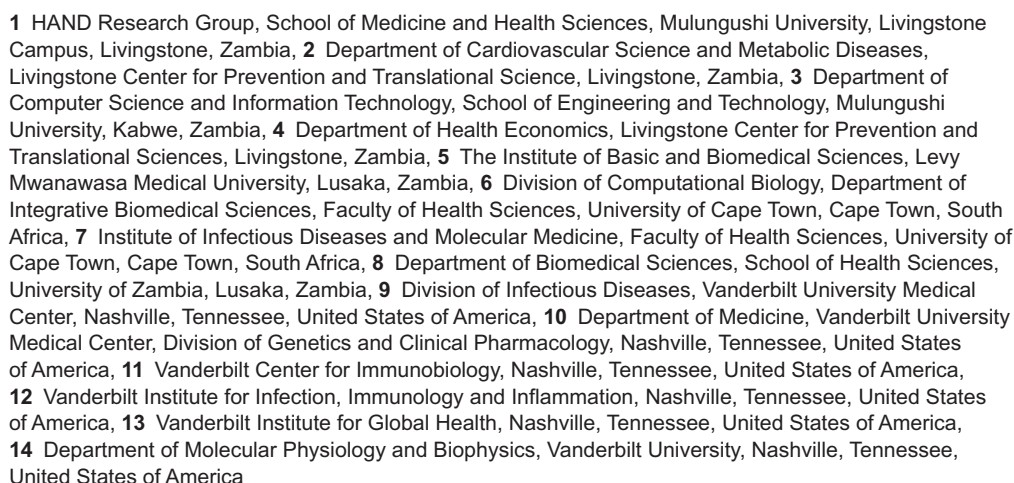

**Data availability statement:** The data are not publicly available due to privacy or ethical restrictions. The data will be made available upon request to qualified researchers, subject to a formal data sharing agreement and approval by our Institutional Review Board, which ensures data will be handled in accordance with the ethical restrictions under which they were collected. To access the data please contact the Vanderbilt University Medical Center Institutional Review Board, 2525 West End Avenue Suite 600, Nashville, TN 37203-8820; Phone +1 (615) 322-2918; erin.johnson@vumc.org or jacqueline.m.van.audenhove@vumc.org.

**Funding:** This work was supported by the Fogarty International Center and National Institute of Diabetes and Digestive and Kidney Diseases of the National Institutes of Health grants (R21TW012635 to SKM and AK), (R01HL147818 and R01HL144941 to AK), National Institute of Allergy and Infectious Diseases grant (K23 AI100700 to JK), National Center for Advancing Translational Sciences grant (UL1 TR002243 to JK), the Tennessee Center for AIDS Research grant (P30 AI110527 to JK), and National Heart, Lung, and Blood Institute grant (K01HL130497 to JK) and the American Heart Association Award (24IVPHA1297559 to SKM and AK). The funders had no role in study design, data collection and analysis, decision to publish, or preparation of the manuscript.

**Competing interests:** The authors have declared that no competing interests exist.

IR. Mitochondrial-metabolically associated factors were dominant: propionylcarnitine (C3uM) ($\beta = 4.727$, $p = 0.004$), isoleucine ($\beta = 0.051$, $p < 0.0001$), glutamic acid ($\beta = 0.035$, $p = 0.0001$), and leptin ($\beta = 0.097$, $p < 0.0001$). Inflammation and dyslipidemia were mild determinants in the adjusted model: interleukin-6 (IL-6)($\beta = 0.274$, $p = 0.0003$), interleukin-17 (IL-17) ($\beta = 0.009$; $p = 0.004$), tumor necrosis factor-alpha (TNF-α)($\beta = 0.337$, $p = 0.001$), monocyte chemoattractant protein-1 (MCP-1)($\beta = 0.003$; $p = 0.005$), fasting TG ($\beta = 0.007$; $p = 0.006$) and leptin ($\beta = 0.041$; $p = 0.009$). Glycine showed an inverse association ($\beta = -0.005$; $p = 0.006$)**.** Two IR phenotypes emerged: metabolic, characterized by impaired branched-chain amino acid catabolism, elevated C3uM levels, which implicates mitochondrial overload and gut-derived metabolic flux as early drivers of IR, and inflammatory-lipotoxic, associated with adipose cytokine release in HIV infection, necessitating distinct biomarkers for early detection and individualized intervention.

## Introduction

Insulin resistance is a key risk factor for metabolic diseases, even in normoglycemic individuals. Beyond obesity, emerging roles of adipokines (leptin, adiponectin), cytokines (IL-6, TNF-α), and amino acids (branched-chain amino acids) are implicated in IR pathogenesis. IR is a fundamental pathophysiological defect underlying type 2 diabetes (T2D) and cardiovascular disease (CVD), even in individuals who have not yet developed overt hyperglycemia [1,2]. While obesity and dyslipidemia are well-established contributors to IR, emerging evidence highlights the critical roles of adipokine dysregulation, chronic low-grade inflammation, and perturbations in amino acid metabolism in its pathogenesis [1,3]. The HOMA2-IR provides a validated and widely used estimate of IR derived from fasting glucose and insulin levels [4]. Yet, comprehensive studies integrating these multifactorial determinants in normoglycemic populations remain scarce.

Adipose tissue dysfunction, characterized by altered secretion of adipokines such as leptin and adiponectin, contributes to systemic metabolic derangements. Leptin, a hormone primarily involved in appetite regulation, has been implicated in promoting IR through pro-inflammatory signaling in adipose tissue, while reduced adiponectin levels correlate with impaired insulin sensitivity [5]. Concurrently, chronic inflammation, marked by elevated cytokines such as interleukin-6 (IL-6) and tumor necrosis factor-alpha (TNF-α), further exacerbates IR by interfering with insulin receptor signaling [6,7]. Furthermore, metabolomic studies have identified branched-chain and aromatic amino acids (leucine, tyrosine, and glutamic acid) as potential biomarkers of IR, possibly reflecting mitochondrial dysfunction or altered gluconeogenic flux [8]. Despite these advances, most prior research has focused on diabetic or high-risk cohorts, leaving a gap in understanding early IR mechanisms in normoglycemic individuals [9].

While there are established links between obesity, dyslipidemia, and IR, critical gaps persist in understanding how integrated pathophysiological pathways contribute

to IR development in normoglycemic individuals [10]. These compartmentalized research findings overlook the interconnected crosstalk in the early pathogenesis of insulin resistance. Further, Methodological variability in insulin resistance assessment, particularly between indirect measures like HOMA-IR and gold-standard clamp techniques, significantly hinders reliable cross-study comparisons of multifactorial contributions.

This study aimed to elucidate the integrated contributions of metabolic, inflammatory, and adipokine-related pathways to IR in normoglycemic adults using HOMA2-IR. We hypothesized that obesity-associated dyslipidemia, chronic inflammation, adipokine imbalance, and amino acid dysmetabolism synergistically drive IR, even in the absence of diabetes. By employing a cross-sectional design with rigorous adjustment for confounders (age, sex, and BMI), we sought to identify independent correlates of IR, providing insights into potential therapeutic targets for early intervention. Our findings may inform strategies to mitigate diabetes and CVD risk by addressing these multifactorial contributors before overt metabolic disease develops.

## Methods

### Ethics statement

This study was approved on September 11, 2012, by the Vanderbilt University Institutional Review Board (IRB# 121175). All participants provided written informed consent. All data were handled with strict adherence to privacy and confidentiality standards, with appropriate de-identification applied to protect participant identity. The authors had no access to information that could identify individual participants during or after data collection. The conduct of the research followed the ethical principles outlined in the Declaration of Helsinki.

### Study design and participants

This was a retrospective cross-sectional analysis of 100 normoglycemic adults aged between 18 and 65 years enrolled from the Vanderbilt Comprehensive Care Clinic. The full methodology has been previously described in detail [11–14]; a brief overview is presented below.

### Inclusion and exclusion criteria

This retrospective cross-sectional study recruited through a census, 100 normoglycemic adults aged 18–65 years, including individuals with and without treated HIV. Among the 70 participants with HIV, all were on antiretroviral therapy for at least two years, with immune reconstitution (CD4 + count >350 cells/µL) and sustained plasma viral suppression for at least six months of enrollment. Waist circumference was used as a pre-screening tool, aiming for balanced representation above and below the US National Cholesterol Education Program Adult Treatment Panel III (NCEP)-defined metabolic syndrome thresholds. 35 participants living with HIV were obese (BMI > 30 kg/m²), and >35 kg/m² where possible, to ensure adequate representation of obesity-related metabolic profiles. Individuals without HIV were recruited from the Vanderbilt general medicine clinics. Exclusion criteria for all participants included pregnancy, post-menopausal status (for women), and current use of anti-diabetic medications or statins.

### Data collection

Data collection for this cohort started on 1st October 2012 until 20th August 2013. Measurements of body mass index (BMI), waist/hip circumference, and blood pressure were standardized [11]. Hypertension was defined as systolic/diastolic BP ≥ 130/85 mmHg or antihypertensive use [15]. Fasting blood samples were analyzed for glucose, insulin, lipids (total cholesterol, LDL, HDL, triglycerides), and glycated haemoglobin A1c (HbA1c). Adipokines (leptin, adiponectin) and inflammatory markers were measured using a multiplex bead assay, and amino acids using Liquid Chromatography–Mass Spectrometry (LC-MS). HOMA2-IR, HOMA2-S (sensitivity), and other IR metrics, HOMA2-B (β-cell function), were

calculated using standard calculator [11,16–18]. Amino acids and acylcarnitines were analyzed using an Agilent 1290 Infinity high-performance liquid chromatography (HPLC) system coupled to a 6490 Triple Quadrupole Mass Spectrometer, operated in multiple reaction monitoring (MRM) mode. All analyses were conducted at Agilent Technologies in Wilmington, DE, ensuring high sensitivity and precision across metabolite classes.

All analyses were performed using a Dionex UltiMate 3000 high-performance liquid chromatography (HPLC) system coupled to a Thermo Scientific Quantiva Triple Quadrupole Mass Spectrometer operating in multiple reaction monitoring (MRM) mode, at Thermo Fisher Scientific in San Jose, CA. Calibration standards were prepared from certified reference materials following established methodologies. For free fatty acid (FFA) quantification, a specialized analytical protocol was employed through the Vanderbilt Lipid Sub-Core facility. The workflow involved plasma extraction using a heptane/isopropanol solvent system, followed by purification via silica gel thin-layer chromatography. Derivatization was then carried out using boron trifluoride in methanol to form methyl esters, which were subsequently quantified on an Agilent 7890 gas chromatograph using authentic reference standards for accuracy.

Fasting blood samples (10 ml) were collected in EDTA-coated vacutainers, promptly centrifuged at 4°C for 10 minutes, and the separated plasma was stored at −80°C for subsequent metabolomic analysis. Fasting plasma glucose and insulin levels were assessed, and insulin resistance was determined using the HOMA2-IR formula, available at www.dtu.ox.ac.uk/homacalculator. Body weight was recorded using a digital scale, height was measured with a wall-mounted stadiometer, and waist circumference was taken with a flexible measuring tape positioned horizontally one inch above the belly button.

Fat mass index (FMI) was derived by dividing total fat mass (kg) by height squared (m²), offering a more precise measure of adiposity than BMI or body fat percentage, as it accounts for the nonlinear association between fat-free mass and height.

## Data analysis plan

Mean values with SD, median values with interquartile ranges (IQR) or percentages were computed for demographic and clinical characteristics, as well as plasma metabolite levels of amino acids, acylcarnitines, organic acids, and free fatty acids, stratified by obesity status.

Normality was assessed using the Shapiro-Wilk test, with a p-value > 0.05 indicating approximate normality. Based on the data distribution, parametric tests were applied to normally distributed variables, while non-parametric tests were used for variables that did not follow a normal distribution. Chi-square tests were used to compare categorical IR and categorical independent variables. Fisher's exact test was used to compare categorical IR and categorical independent variables where over 20% of cells had an expected count less than 5. For the association of HOMA2-IR, a continuous outcome variable, Pearson's correlation assessed linear relationships between normally distributed variables. Spearman's rank correlation evaluated monotonic associations for non-parametric data. A linear regression modelled correlations between HOMA2-IR with all continuous variables, was used. For the multivariable linear regression analyses, covariates were selected a priori based on their established biological and clinical relevance to insulin resistance, and all prespecified covariates were entered into the model in a stepwise selection.

HOMA2-IR, initially calculated as a continuous quantitative measure of insulin resistance, was transformed into a dichotomous variable to classify participants into "insulin resistant" versus "non–insulin resistant" groups. To accomplish this, we applied a predefined threshold of HOMA2-IR 1.9, which is consistent with published epidemiological and metabolic research identifying early insulin resistance. This value as indicative of clinically meaningful insulin resistance. Values above 1.9 cutoff were coded as "1" (elevated insulin resistance), and values at or below the cutoff were coded as "0" [19]. This approach allowed for categorical comparisons while maintaining alignment with established literature and biologically relevant definitions.

The relationship between plasma metabolites and HOMA2-IR was assessed using multivariable linear regression, adjusting for age, sex, and BMI. The analyses were performed using StatCrunch and GraphPad Prism. A two-tailed p-value <0.05 defined statistical significance.

We used the Strengthening the Reporting of Observational Studies in Epidemiology to guide the writing (S1 File).

## Results

### Participant characteristics and prevalence of insulin resistance

Our cohort of 100 normoglycemic adults had a median age of 44 years (IQR: 35–49), with a high prevalence of obesity (65%) and hypertension (37%). The overall prevalence of insulin resistance (IR), defined by HOMA2-IR, was 36% (36/100). The distribution of IR was significantly associated with clinical phenotypes: the vast majority of individuals with IR were obese, and IR was more common in hypertensive participants compared to normotensive individuals. A significant difference was also observed based on HIV status, with a higher prevalence of IR in HIV-negative participants compared to their HIV-positive counterparts. There was no significant difference in IR prevalence between males and females (Table 1).

### Bivariate associations of HOMA2-IR with demographic, metabolic and inflammatory markers

We first compared the median levels of demographic, anthropometric, metabolic, and inflammatory variables between participants with and without IR. Individuals with IR had significantly larger hip and waist circumferences, higher BMI, higher fasting triglycerides (TG), and lower fasting HDL cholesterol (all $p < 0.01$). As expected, they also exhibited significantly higher HOMA2-β cell function and lower HOMA2 insulin sensitivity (both $p < 0.0001$). There were no significant differences in age, HbA1c, LDL, or total cholesterol between the groups (S1 Fig).

The inflammatory profile markedly differed between groups. Participants with IR had significantly elevated levels of key pro-inflammatory cytokines, including IL-6, and soluble receptors for TNF-α. They also exhibited higher levels of macrophage inflammatory protein-1α, C-reactive protein, and high-sensitivity CRP (S2 Fig). Analysis of amino acid profiles revealed strong associations with IR. Circulating levels of branched-chain amino acids (BCAAs) leucine and isoleucine, as well as the aromatic amino acids tyrosine and tryptophan, were significantly elevated in the IR group. Alanine and glutamic acid were also higher, while glycine, asparagine, and citrulline were significantly lower in individuals with IR (S3 Fig). Several metabolites from organic acid and acylcarnitine pathways were also dysregulated. The IR group had higher levels

**Table 1. Distribution of insulin resistance by gender, hypertension and obesity.**

| Variables | Frequency (%)/Median (IQR)/ Mean (SD) | Insulin resistance YES | Insulin resistance NO | P value |
|---|---|---|---|---|
| **Sex,** *n = 100* | | | | 0.342 |
| *Male* | 52 (52) | 21 (40.4%) | 31 (59.6%) | |
| *Female* | 48 (48) | 15 (31.3%) | 33 (68.8%) | |
| **Hypertension** | | | | **0.043** |
| *Hypertensive* | 37 | 18 (48.7%) | 19 (51.4%) | |
| *Normotensive* | 63 | 18 (28.6%) | 45 (71.4%) | |
| **Obesity** | | | | **<0.0001** |
| *Obese* | 65 | 33 (50.8%) | 32 (49.2%) | |
| *Non-Obese* | 35 | 3 (8.6%) | 32 (91.4%) | |
| **HIV status** | | | | **0.001** |
| *Negative* | 30 | 18 (60%) | 12 (40%) | |
| *Positive* | 70 | 18 (25.7%) | 52 (74.3%) | |

**HIV;** Human immunodeficiency virus.

of lactate, pyruvate, and the acylcarnitines C4-butyryl, C5-isovaleryl, and C5-methylbutyryl. The ratio of C2/C3/C5 acylcarnitines was significantly lower in the IR group (**S4 Fig**).

### Linear regression analyses of factors associated with HOMA2-IR

Simple linear regression analyses confirmed the unadjusted associations between HOMA2-IR and the variables described above. HOMA2-IR was positively correlated with adiposity measures (hip and waist circumference), lipids (TG, total cholesterol), glycemic indices (HbA1c, glucose), HOMA2-β, and numerous inflammatory cytokines and amino acids. It was inversely correlated with HDL and HOMA2-S (**Fig 1**).

A similar series of simple linear regressions detailed the positive associations between HOMA2-IR and inflammatory markers such as IL-6, TNF-α, IL-17, and MCP-1, among others (**Fig 2**).

The positive univariate relationships between HOMA2-IR and metabolites like lactate, pyruvate, and specific acylcarnitines (C3, C5-isovaleryl) were further detailed (**Fig 3**).

Finally, the unadjusted associations of amino acids with HOMA2-IR were plotted, visually reinforcing the positive relationships with BCAAs and glutamic acid and the inverse relationship with glycine (**Fig 4**).

### Multivariable model of independent predictors of HOMA2-IR

In the multivariable linear regression model adjusted for key covariates, several factors retained strong, independent associations with HOMA2-IR (**Table 2**). The model explained a substantial proportion of the variance in IR ($R^2 = 0.94$). Leptin ($\beta = 0.041$, $p = 0.009$) and fasting triglycerides ($\beta = 0.007$, $p = 0.006$) were significant independent predictors. Among amino acids, glutamic acid ($\beta = 0.025$, $p = 0.0001$) and tyrosine ($\beta = 0.029$, $p = 0.0004$) showed strong positive associations, while glycine was inversely associated ($\beta = -0.005$, $p = 0.006$). The inflammatory markers IL-17 ($\beta = 0.009$, $p = 0.004$) and MCP-1 ($\beta = 0.003$, $p = 0.005$) were also independent correlates. Notably, the mitochondrial metabolism-associated metabolite propionylcarnitine (C3) remained a significant predictor ($\beta = 2.811$, $p = 0.034$).

To illustrate the effect sizes and precision of these key multivariate associations, a forest plot was constructed (**S5 Fig**). This plot summarizes the adjusted beta coefficients and 95% confidence intervals for selected metabolic and inflammatory markers, stratified by obesity status, providing a clear visual representation of their relative contributions to IR.

## Discussion

### Summary of main findings

It is important to emphasize that the present study was conducted in a population predominantly composed of PLWH, who represented approximately 70% of the sample. As such, the observed metabolic and inflammatory patterns must be interpreted within the unique pathophysiological context of chronic HIV infection and long-term ART. These factors introduce biological heterogeneity not fully comparable to normoglycemic or HIV-uninfected populations, and therefore the generalizability of our findings outside HIV-endemic settings should be considered with caution.

This study provides compelling evidence that IR in normoglycemic adults, with 70% of the cohort living with HIV, manifests through two mechanistically distinct pathways: a metabolic phenotype and an inflammatory-lipotoxic phenotype. The study demonstrated that elevations in BCAAs, aromatic amino acids, and C3uM are associated with IR, suggesting mitochondrial overload and amino acid dysmetabolism as notable contributors. The study goes further to reveal that the markers of systemic inflammation (TNF-α, IL-6, IL-17), dyslipidemia (elevated triglycerides and low HDL), and endothelial activation (VCAM-1) were independently associated with IR, reaffirming the central role of adipose tissue–mediated inflammation and lipid toxicity in metabolic dysfunction. Notably, glycine consistently demonstrated an inverse association with IR, underscoring its potential preventive role.

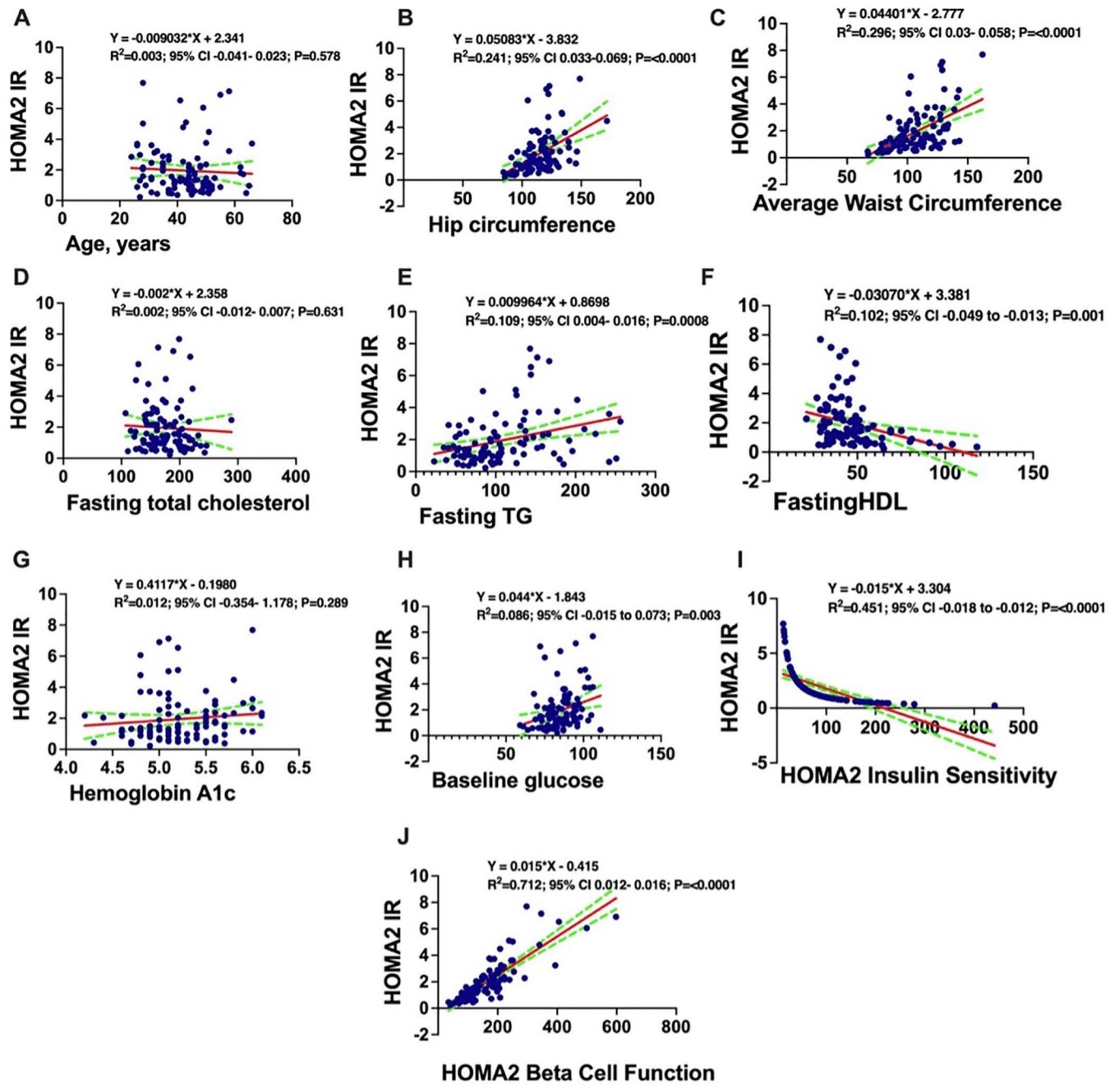

**Fig 1. Simple linear regression analysis demonstrating association between sociodemographic, lipogram and HOMA2-IR.** HOMA2-IR absolute values were positively associated with Age, years **(A)**, Hip circumference **(B)**, Average waist circumference **(C)**, fasting total cholesterol **(D)**, fasting TG **(E)**, Hemoglobin A1c **(G)**, Baseline glucose **(H)**, HOMA2 Beta Cell function **(J)**, and negatively associated with fasting HDL **(F)** and HOMA2 Insulin sensitivity **(I)**. y = HOMA2-IR. **HOMAIR;** Homeostatic Model Assessment of Insulin Resistance, **HOMA2;** Homeostatic Model Assessment (version 2), **HDL;** High-Density Lipoprotein, **LDL;** Low-Density Lipoprotein, **TG;** Triglycerides, **TNF-α;** Tumour Necrosis Factor-alpha.

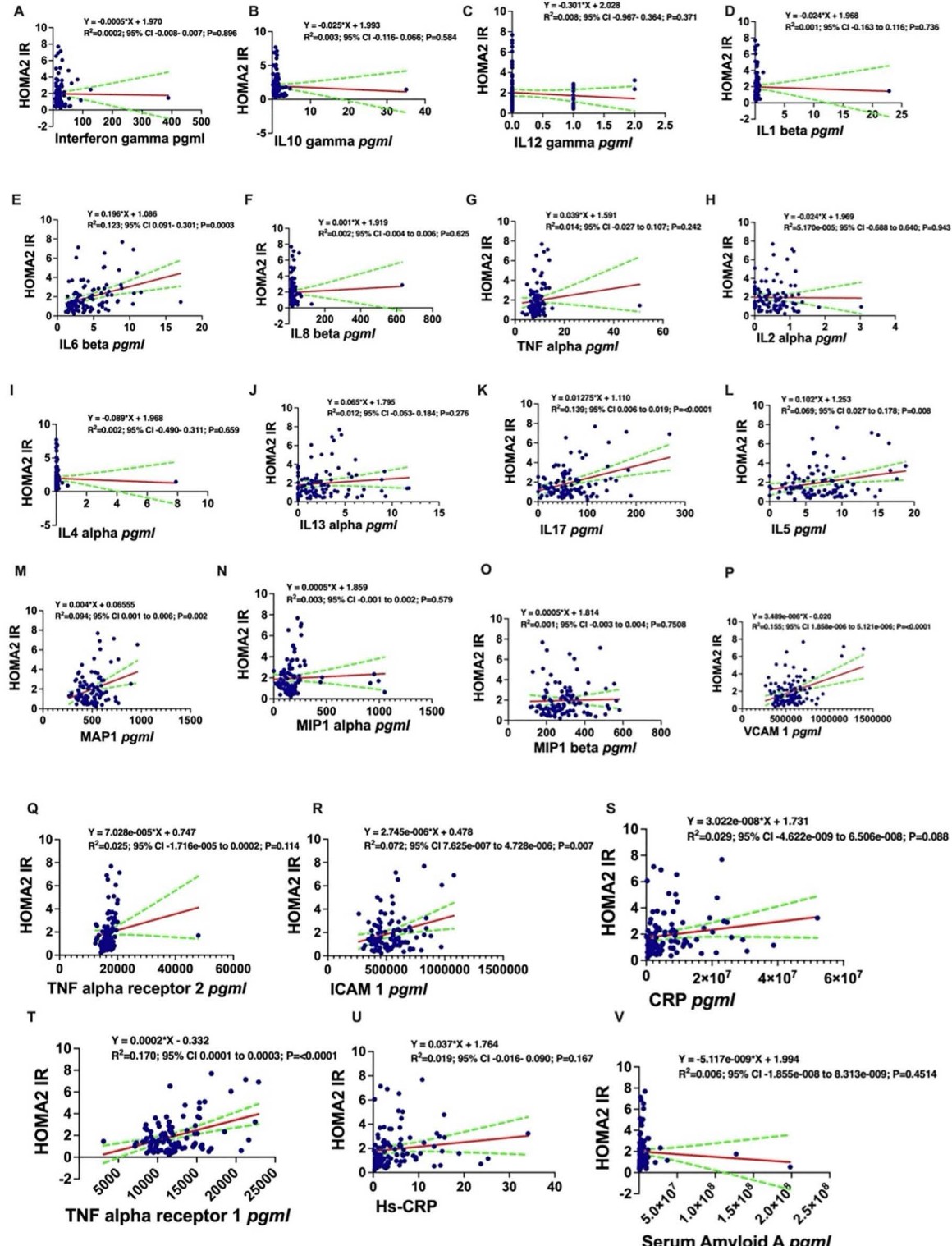

**Fig 2. Simple linear regression analysis demonstrating association between specific inflammatory marker and HOMA2-IR.** HOMA2 IR absolute values were positively associated with IL6 beta **(E)**, IL8 beta **(F)**, TNF alpha **(G)**, IL2 alpha **(H)**, IL13 alpha **(J)**, IL17 **(K)**, IL5 **(L)**, MAP 1(M), MIP1 alpha **(N)**, MIP1 beta **(O)**, VCAM1 **(P)**, TNF alpha receptor 2 **(Q)**, ICAM1 **(R)**, CRP **(S)**, TNF alpha receptor 1 **(T)**, Hs-CRP **(U)** and negatively associated with

Interferon gamma **(A)**, IL10 gamma **(B)**, IL12 gamma **(C)**, IL1 beta **(D)**, IL4 alpha **(I)** and serum Amyloid A **(V)**. y = HOMA2-IR. **TNF-α;** Tumor Necrosis Factor-alpha, **HOMAIR;** Homeostatic Model Assessment of Insulin Resistance, **IL5;** Interleukin-5, **IL6;** Interleukin-6, **IL10;** Interleukin-10, **IL7;** Interleukin-7, MAP; Macrophage chemotactic protein, MIP; Macrophage Inflammatory protein, VCAM1; Vascular adhesion molecule1, ICAM1; Intercellular adhesion molecule CRP; C-Reactive protein, Hs-CRP; High Sensitive_C-Reactive protein.

## Interpretation and connection to literature

The study findings reveal that low-grade inflammation is implicated in the predisposition to IR. The elevation of IL-17 links inflammation to metabolic dysregulation. These findings are similar to a study by Chehimi *et al,* where they revealed that IL-17 promotes adipose tissue inflammation and impairs insulin signaling [20]. This supports the immune–metabolic axis in IR.

This data corroborates well-established pathways involving chronic inflammation, lipid-induced insulin signaling disruption, and endothelial dysfunction. This aligns with the models proposed by Shoelson *et al.* and ter Horst *et al.* emphasizing macrophage infiltration, cytokine overexpression, and lipid metabolite accumulation as primary drivers of IR in HIV infection [21–23].

The study further revealed that IR was also characterized by systemic low-grade inflammation driven by adipose tissue macrophage infiltration in HIV infection, findings that Shikuma *et al.* similarly suggested by [24,25]. IL-6 and TNF-α are central to this inflammatory axis, with MCP-1 playing a key role in monocyte recruitment via the CC-chemokine receptor 2 (Ccr2) receptor [26]. These macrophages propagate local inflammation and interfere with insulin receptor signaling, thus perpetuating resistance [22,27]. Supporting evidence from post-bariatric cohorts demonstrates that a reduction in ATM density correlates with HOMA-IR improvements [28,29].

Fasting TG, IL6, TNF-α and IL17 were among the positively associated factors in the multivariable regression analysis, consistent with findings from Shi *et al.,* who demonstrated that pro-inflammatory cytokines inhibit insulin signaling via transcriptional regulation of miR-146b [30]. Elevated triglycerides and suppressed HDL levels reinforced the lipotoxicity narrative, where diacylglycerol accumulation in hepatocytes activates protein kinase C epsilon (PKCε), reducing insulin receptor tyrosine kinase activity [21].

Although some of our findings mirror patterns reported in the general population, the metabolic phenotype observed here is substantially shaped by HIV-related immune activation, residual inflammation despite viral suppression, and ART-associated alterations in lipid and glucose handling. These HIV-specific mechanisms may amplify traditional cardiometabolic pathways, and therefore direct comparisons to HIV-negative cohorts must be interpreted within this context.

Our findings further reinforce and expand upon prior studies identifying BCAAs and their catabolic intermediates as early metabolic disruptors leading to IR. Lackey *et al.,* Newgard *et al.*, and other studies [31,32] previously reported similar associations between amino acid dysregulation and insulin-resistant phenotypes. This is consistent with the work of Lackey *et al.*, who demonstrated a significant association between elevated plasma levels of BCAAs and insulin-resistant obesity in humans. This underscores the role of amino-acid dysregulation in the pathophysiology of metabolic dysfunction [33], while also acknowledging that reverse causality may contribute to the observed associations, that is, IR could directly increase gluconeogenesis, supported by enhanced protein turnover [34,35]. In a study conducted by She *et al.*, elevated plasma leucine, isoleucine, valine, tyrosine, glutamic acid, and the acylcarnitine C3uM [36], were found to be associated with the dysregulation of key mitochondrial enzymes involved in BCAA metabolism, suggesting a mechanistic link between amino acid accumulation and impaired metabolic processing [37].

Elevated BCAA levels point to inefficient mitochondrial oxidative metabolism, specifically reduced activity of the branched-chain α-keto acid dehydrogenase (BCKDH) complex, which leads to accumulation of metabolic intermediates such as 3-hydroxyisobutyrate. This intermediate has been shown to enhance fatty acid uptake and lipid deposition in skeletal muscle, ultimately impairing insulin signaling through mammalian target of rapamycin complex 1 (mTORC1) overactivation and IRS-1 serine phosphorylation. These mechanisms, as highlighted in the work of Elrayess *et al.,* and She *et al.,* form a plausible explanation for our observed associations [37,38]. The role of C3uM as a robust biomarker of IR further aligns with recent evidence implicating incomplete fatty acid oxidation and gut-derived propionate overflow in subclinical metabolic dysfunction.

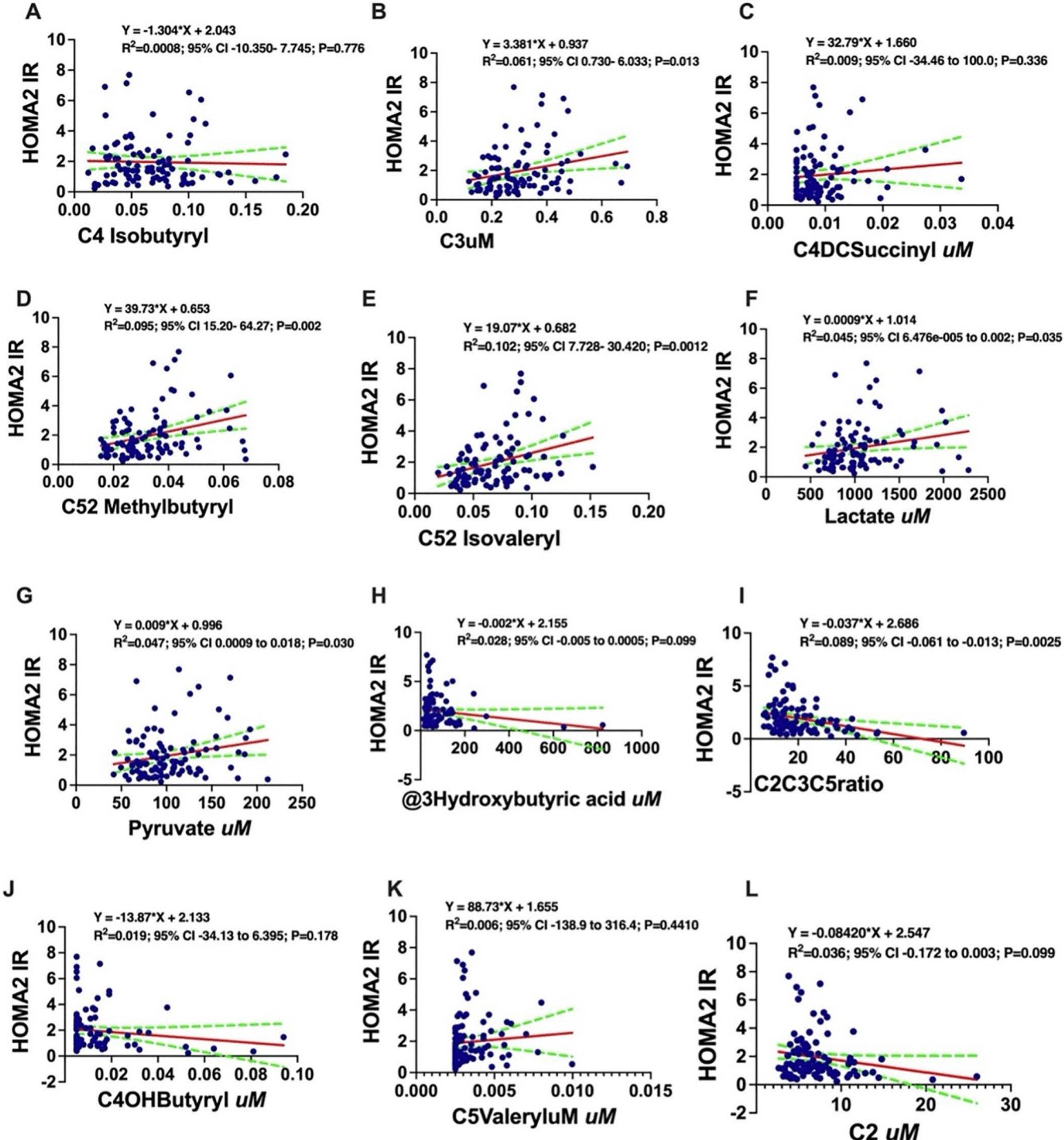

**Fig 3. Simple linear regression analysis demonstrating association between specific *organic acids and acylcarnitine* and HOMA2-IR.**
HOMA2-IR absolute values were positively associated with C4 Isobutryl **(A)**, C3 **(B)**, C4DCSuccinyl **(C)**, C52 Methylbutyryl **(D)**, C5 Isovaleryl **(E)**, Lactate **(F)**, Pyruvate **(G)**, C5Valeryl **(K)** and negatively associated with 3hydroxybutric acid **(H)**, C2C3C5ratio **(I)**, C40HButyryl **(J)** and C2 **(L)**. y = HOMA2-IR.
**HOMAIR;** Homeostatic Model Assessment of Insulin Resistance, **C52;** methylbutyryl, **C5;** Complement Component 5, **C2uM;** Acetylcarnitine, **C2C3C5ratio;** Acylcarnitine Ratio, **C3uM;** Propionylcarnitine.

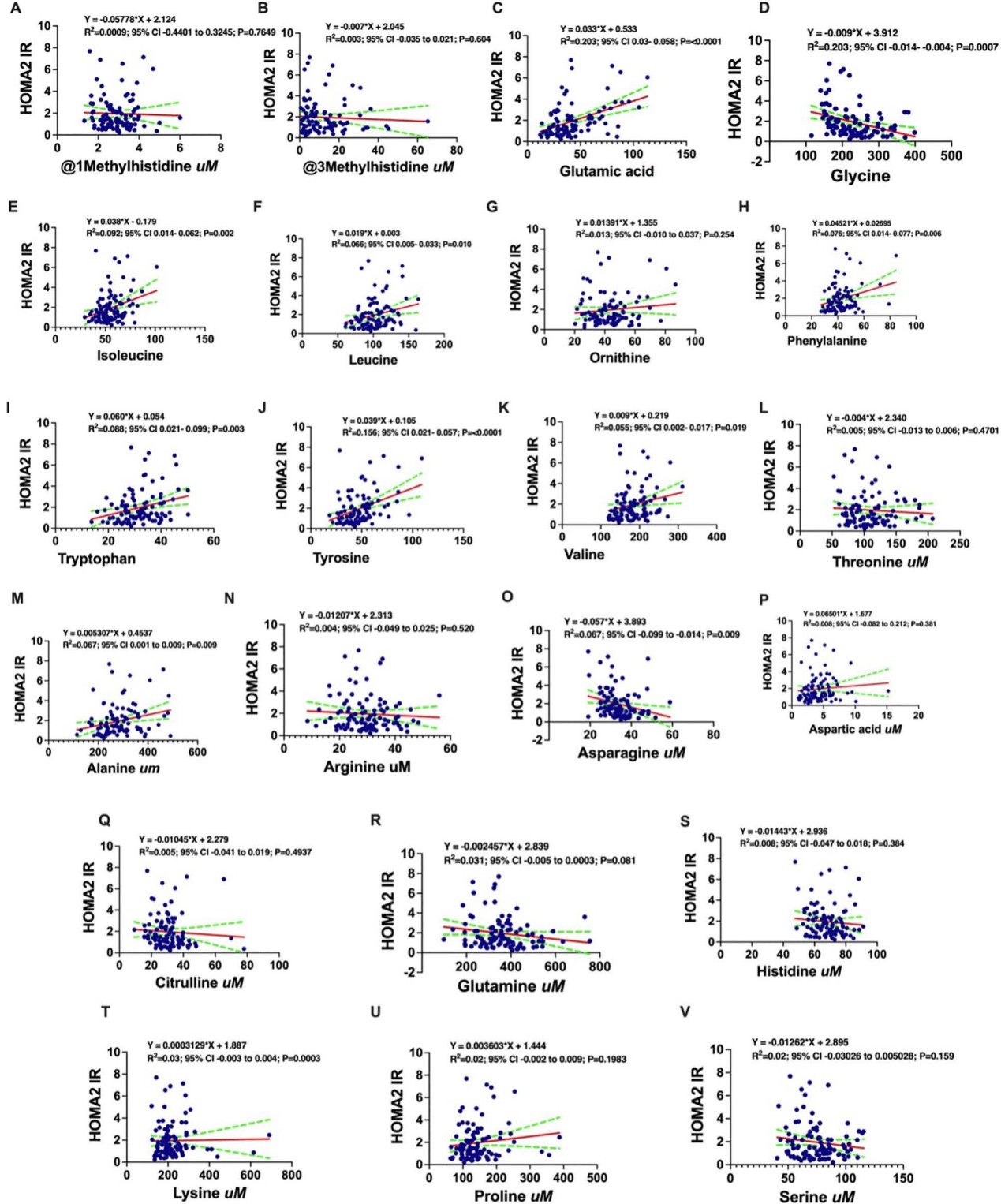

**Fig 4. Simple linear regression analysis demonstrating association between specific amino acids and HOMA2-IR.** HOMA2-IR absolute values were positively associated with Glutamic acid **(C)**, Isoleucine **(E)**, Leucine **(F)**, IL2 alpha **(H)**, VCAM1 **(P)**, TNF alpha receptor 2 **(Q)**, ICAM1 **(R)**, CRP **(S)**, TNF alpha receptor 1 **(T)**, Hs-CRP **(G)**, Phenylalanine **(H)**, Tryptophan **(I)**, Tyrosine **(J)**, Valine **(K)**, Alanine **(M)**, Aspartic acid **(P)**, Proline **(U)** and

negatively associated with 1MethylHistidine **(A)**, 3MethylHistidine **(B)**, Glycine **(D)**, Threonine **(L)**, Citrulline **(Q)**, Glutamine **(R)**, Histidine **(S)** and Serine **(V)**. y = HOMA2-IR. **HDL;** High-Density Lipoprotein, **TG;** Triglycerides, **TNF-α;** Tumor Necrosis Factor-alpha, **HOMAIR;** Homeostatic Model Assessment of Insulin Resistance, **HOMA2;** Homeostatic Model Assessment (version 2), **IL5;** Interleukin-5, **IL6;** Interleukin-6, **IL10;** Interleukin-10, **IL7;** Interleukin-7, **C52;** methylbutyryl, **C5;** Complement Component 5, **C2uM;** Acetylcarnitine, **C2C3C5ratio;** Acylcarnitine Ratio, **C3uM;** Propionylcarnitine.

**Table 2. Factors associated with insulin resistance.**

| Variables | Univariate linear | | Multivariate linear | | |
|---|---|---|---|---|---|
| | β (95%CI) | P value | β* (95%CI) | Adj. R² | P value |
| **Fasting HDL** | −0.031 (−0.049, −0.012) | **0.001** | −0.014 (−0.032, 0.003) | 0.315 | 0.099 |
| **Fasting TG** | 0.01 (0.004, 0.016) | **0.001** | 0.007 (0.002, 0.012) | 0.351 | **0.006** |
| **Leptin *ng/ml*** | 0.042 (0.025, 0.059) | **<0.0001** | 0.041 (0.011, 0.072) | 0.344 | **0.009** |
| **Adiponectin *pgml*** | 0.000 (0.000, 0.000) | **0.016** | −0.000 (−3.935, 2.141) | 0.298 | 0.559 |
| **Baseline glucose** | 0.044 (0.015, 0.073) | **0.003** | 0.026 (0.0002, 0.052) | 0.324 | **0.048** |
| **Baseline insulin *uUml*** | 0.119 (0.117, 0.120) | **<0.0001** | 0.117 (0.115, 0.119) | 0.996 | **<0.0001** |
| **@90-minute glucose** | 0.014 (0.006, 0.022) | **0.001** | 0.009 (0.003, 0.017) | 0.351 | **0.006** |
| **@90-minute insulin *uUml*** | 0.013 (0.010, 0.016) | **<0.0001** | 0.011 (0.008, 0.018) | 0.558 | **<0.0001** |
| **@120-minute glucose** | 0.019 (0.011, 0.028) | **<0.0001** | 0.014 (0.007, 0.022) | 0.386 | **0.0003** |
| **Baseline HOMAIR** | 0.517 (0.503, 0.532) | **<0.0001** | 0.519 (0.501, 0.537) | 0.980 | **<0.0001** |
| **HOMA2 insulin sensitivity S** | −0.015 (−0.019, −0.012) | **<0.0001** | −0.012 (−0.016, −0.008) | 0.487 | **<0.0001** |
| **HOMA2 beta cell function B** | 0.015 (0.013, 0.016) | **<0.0001** | 0.013 (0.011, 0.015) | 0.752 | **<0.0001** |
| **Averaged hip circumference *cm*** | 0.051 (0.033, 0.069) | **<0.0001** | −0.011 (−0.062, 0.039) | 0.297 | 0.991 |
| **Averaged waist circumference *cm*** | 0.044 (0.033, 0.069) | **<0.0001** | 0.016 (−0.018, 0.049) | 0.302 | 0.774 |
| **IL6 *pgml*** | 0.196 (0.091, 0.301) | **<0.0001** | 0.068 (−0.044, 0.180) | 0.306 | 0.232 |
| **IL17 *pgml*** | 0.013 (0.006, 0.019) | **<0.0001** | 0.009 (0.003, 0.014) | 0.356 | **0.004** |
| **IL5 *pgml*** | 0.102 (0.027, 0.178) | **0.008** | 0.061 (−0.011, 0.132) | 0.316 | 0.095 |
| **Macrophage chemotactic protein 1 *pgml*** | 0.003 (0.001, 0.006) | **0.002** | 0.003 (0.001, 0.005) | 0.351 | **0.005** |
| **Alanine *uM*** | 0.005 (0.001, 0.009) | **0.009** | 0.002 (−0.001, 0.006) | 0.309 | 0.177 |
| **Asparagine *uM*** | −0.057 (−0.100, −0.014) | **0.009** | −0.025 (−0.063, 0.014) | 0.307 | 0.207 |
| **Glutamic acid *uM*** | 0.033 (0.020, 0.046) | **<0.0001** | 0.025 (0.013, 0.037) | 0.397 | **0.0001** |
| **Glycine *uM*** | −0.009 (−0.014, −0.004) | **0.001** | −0.005 (−0.01, −0.001) | 0.348 | **0.006** |
| **Leucine *uM*** | 0.019 (0.005, 0.033) | **0.010** | 0.013 (−0.001, 0.026) | 0.320 | 0.068 |
| **Phenylalanine *uM*** | 0.045 (0.014, 0.077) | **0.006** | 0.028 (−0.0001, 0.056) | 0.324 | 0.05 |
| **Tryptophan *uM*** | 0.060 (0.021, 0.099) | **0.003** | 0.038 (−0.002, 0.073) | 0.327 | **0.037** |
| **Tyrosine *uM*** | 0.039 (0.021, 0.057) | **<0.0001** | 0.029 (0.013, 0.045) | 0.384 | **0.0004** |
| **Valine *uM*** | 0.009 (0.002, 0.017) | **0.019** | 0.005 (−0.002, 0.012) | 0.310 | 0.158 |
| **C52 Methylbutyryl *uM*** | 39.736 (15.221, 64.284) | **0.002** | 22.819 (−1.181, 46.821) | 0.321 | 0.0621 |
| **C5 Isovaleryl *uM*** | 19.075 (7.745, 30.433) | **0.001** | 11.514 (0.179, 22.849) | 0.324 | **0.047** |
| **Lactate *uM*** | 0.001 (0.000, 0.002) | **0.035** | 0.001 (−0.0001, 0.0001) | 0.316 | 0.092 |
| **Pyruvate *uM*** | 0.009 (0.001, 0.018) | **0.030** | 0.005 (−0.002, 0.013) | 0.311 | 0.146 |
| **C2uM** | −0.084 (−0.171, 0.003) | 0.059 | −0.055 (−0.132, 0.021) | 0.310 | 0.153 |
| **C2C3C5ratio** | −0.037 (−0.061, −0.013) | **0.003** | −0.029 (−0.052, −0.005) | 0.336 | **0.017** |
| **C3uM** | 3.36 (0.728, 6.031) | **0.013** | 2.811 (0.215, 5.406) | 0.328 | **0.034** |

**HDL;** High-Density Lipoprotein, **TG;** Triglycerides, **TNF-α;** Tumor Necrosis Factor-alpha, **HOMAIR;** Homeostatic Model Assessment of Insulin Resistance, **HOMA2;** Homeostatic Model Assessment (version 2), **IL5;** Interleukin-5, **IL6;** Interleukin-6, **IL10;** Interleukin-10, **IL7;** Interleukin-7, **C52;** methylbutyryl, **C5;** Complement Component 5, **C2uM;** Acetylcarnitine, **C2C3C5ratio;** Acylcarnitine Ratio, **C3uM;** Propionylcarnitine.

## Significance and Implications

The interaction between HIV infection, ART exposure, and classical cardiometabolic factors creates a compounded metabolic burden not typically seen in normoglycemic, HIV-uninfected populations. Visceral adiposity and dyslipidemia in PLWH often arise in the setting of ART-induced adipocyte dysfunction [39], mitochondrial stress, and altered fat redistribution patterns [40]. This may explain why the magnitude of association between central obesity, dyslipidemia, and insulin resistance was stronger in our cohort than reported in HIV-negative populations.

These findings hold considerable translational value within the population of PLWH. First, they support the emerging paradigm that insulin resistance is not exclusively a consequence of obesity but may also arise from, HIV infection mitochondrial inefficiency and amino acid dysregulation in normal-weight individuals, a concept akin to the "metabolically obese, normal weight" (MONW) phenotype [41]. This has implications for clinical screening, where body mass index alone may miss high-risk individuals.

Second, the identification of C3uM and glycine as phenotype-specific biomarkers opens the door for more precise risk stratification and therapeutic targeting. For instance, the metabolically-dysregulated IR phenotype may be responsive to interventions aimed at improving mitochondrial flexibility, such as BCAA restriction, glycine supplementation, and AMPK activation, whereas the obesity-linked IR phenotype, they may benefit more from anti-inflammatory strategies, visceral fat reduction, and Glucagon-like peptide-1 (GLP-1) receptor agonists [20,42]. These insights can inform future public health policies and precision medicine approaches in managing prediabetes.

## Positioning in the global literature

This study complements emerging data from global cohorts, such as the Framingham Offspring Study, KORA (Germany), and the NHANES metabolomics cohort, which have linked BCAAs, acylcarnitines, and inflammatory cytokines to pre-diabetic insulin resistance. However, it advances the field by introducing novel dimensions in three key areas: first, through modelling in a fully normoglycemic population. Few studies have comprehensively dissected IR drivers with stratified modelling in a population of all normoglycemic participants, providing phenotype-specific insights rarely achieved in cross-sectional work, contributing rare mechanistic insight from a context experiencing rapid epidemiological transition.

Second, it highlights the significance of C3uM, a less commonly studied acylcarnitine, as a biomarker of mitochondrial dysfunction and metabolic inflexibility, thereby expanding the current BCAA-focused narrative and bridging metabolomic findings.

Our findings validate the concept of insulin resistance subtypes, akin to Ahlqvist *et al.'s* diabetes clustering, where an attempt was made to disentangle the heterogeneity of diabetes mellitus progression [43,44]. IR marked by mitochondrial amino acid stress and early vascular dysfunction may benefit from interventions targeting metabolic flexibility through BCAA restriction, glycine supplementation, or metformin-induced AMPK activation [45,46]. The obese IR phenotype may require anti-inflammatory therapies, visceral fat reduction, and endothelial-protective strategies such as GLP-1 agonists [47], or Peroxisome proliferator-activated receptor gamma (PPARγ) modulators [48,49].

Furthermore, the elevation of C3uM and depletion of glycine suggest novel biomarkers that could be developed into point-of-care tools for prediabetes screening, particularly in low-resource settings.

Glycine's consistent inverse relationship with IR across phenotypes underscores its dual biological relevance. As a glutathione precursor, it strengthens mitochondrial antioxidant defences and attenuates reactive oxygen species (ROS)-mediated IR [50,51]. Simultaneously, glycine serves as a ligand for (Glycine receptors) GlyRs in adipocytes, where it suppresses catecholamine-stimulated lipolysis [52] and limits free fatty acid spillover, an underappreciated mechanism in adipose IR [53]. This positions glycine supplementation as a novel therapeutic strategy for slowing the progression of insulin resistance and potentially delaying the onset of type 2 diabetes mellitus.

Overall, while our findings provide critical insights into the metabolic and inflammatory drivers of insulin resistance, their interpretation must remain grounded in the HIV-specific physiological context of the studied population. Future research

comparing PLWH and HIV-negative individuals in parallel will be essential for disentangling HIV-related mechanisms from those observed in the general population.

## Strengths and limitations

Despite its cross-sectional design, this study introduces a robust, stratified analytical framework that distinguishes distinct IR phenotypes in a normoglycemic population. The integration of high-resolution metabolomic and cytokine profiling with regression models adjusted for age and sex enabled the detection of subtle but clinically significant metabolic signatures. The identification of C3uM as a specific IR biomarker and the consistent protective signal of glycine across phenotypes underscore the study's novelty and potential clinical utility.

   A key limitation of the study is that the findings derive from a sample with a high burden of HIV infection, which restricts applicability to the general population. The dominant influence of HIV-related inflammation and ART-mediated metabolic changes means these results primarily reflect the cardiometabolic physiology of PLWH rather than normoglycemic, HIV-uninfected individuals [54,55]. Furthering the limitations of this study, it is important to note that, the cross-sectional nature of the study precludes causal inferences regarding the temporal evolution of IR and biomarker changes. Although adequately powered for primary analyses, the sample size may have constrained subgroup analyses, particularly among non-obese individuals. Furthermore, Although HOMA-based indices are widely used and practical for large epidemiological studies, their validity is not uniform across populations or metabolic phenotypes. Prior work has demonstrated reduced concordance between HOMA-IR and hyperinsulinemic–euglycemic clamp measurements in certain ethnic groups, and in individuals with altered β-cell function or disproportionate hepatic versus peripheral insulin resistance [56,57]. Consequently, our findings should be interpreted with the understanding that HOMA2-IR provides an indirect approximation rather than a dynamic, mechanistic assessment of insulin action. Future studies incorporating gold-standard clamp techniques or mixed-meal/OGTT-based modelling would offer more granular insights and validate the patterns observed in this cohort. Finally, unmeasured variables such as dietary intake, physical activity, and hormonal influences may have introduced residual confounding.

## Conclusion

This study delineates two mechanistically distinct IR phenotypes in a normoglycemic but high–HIV-burden cohort. The first is a metabolic archetype characterized by impaired branched-chain amino acid catabolism and elevated C3uM levels, suggesting early mitochondrial stress and altered gut-derived metabolic flux drivers of IR. The second was an inflammatory-lipotoxic archetype, defined by adipose tissue–mediated cytokine release, consistent with canonical models of IR. Metabolic markers such as C3uM, in combination with inflammatory and adipokine profiles, may support earlier detection of subclinical insulin resistance before dysglycemia develops. These phenotype-specific biomarker panels could improve metabolic risk stratification, inform targeted preventive strategies, and refine prediabetes screening in settings with a high HIV burden. Future research should validate these signatures in larger and more diverse cohorts, including HIV-uninfected populations, to determine the extent to which the observed mechanisms are HIV-specific or generalizable. Development of a rapid, point-of-care C3uM assay may further support clinical translation and integration into routine metabolic risk assessment and early diabetes prevention.

### What is Known

- Prior literature has identified obesity, dyslipidemia, chronic low-grade inflammation, and adipokine imbalance as major contributors to IR.

- More recently, metabolomic investigations have revealed consistent associations between elevated BCAAs such as isoleucine and valine and IR, implicating mitochondrial dysfunction in the regulation of substrate oxidation and gluconeogenic flux.

- Leptin resistance and low adiponectin levels have emerged as pivotal mediators of adipose tissue dysfunction. Similarly, inflammatory cytokines such as IL-6 and TNF-α are known to disrupt insulin signaling pathways.

- Tools such as HOMA2-IR offer validated means to estimate insulin resistance using fasting glucose and insulin concentrations, enabling deeper metabolic phenotyping in at-risk yet undiagnosed populations.

**What is Not Known**

- Despite advances in our understanding of insulin resistance, several key gaps remain. First, most studies examine isolated biological pathways, such as lipid metabolism, inflammation, or amino acid dysregulation, without integrating them to evaluate their combined or synergistic roles in early IR pathogenesis.

- Second, mechanistic insights into how mitochondrial metabolites like propionylcarnitine and amino acid flux differentially contribute to IR in non-obese versus obese individuals remain poorly delineated.

- Third, while inflammation and dyslipidemia are recognized as dominant drivers in obesity-related IR, the existence of a distinct mitochondrial-metabolic IR phenotype in non-obese individuals has not been thoroughly validated in African or HIV-treated populations.

- Clinical utility of novel metabolic biomarkers like C3uM as early indicators of IR, especially in the absence of hyperglycemia, remains to be fully explored.

## Supporting information

**S1 File. Strobe.**
(DOCX)

**S1 Fig. Characteristics of insulin resistance by lipids, glycemic indices, demographic & anthropometric factors between people with and without IR.** This figure shows the median (interquartile range, IQR) between people with and without IR, respectively: A) Age, 39 (31.00, 48.75) vs 44.5 (38.25, 49.00)/years, p = 0.074, B) Fasting HDL, 40 (33.50, 45.00) vs 47.5 (36.25, 56.75)/, mg/dL p = 0.0023, C) Hip circumference, 120.2(115.8, 131.1) vs 107.0(98.25, 119.2)/cm, p = <0.0001 D) Waist circumference, 125.2 (104.0, 130.5) vs 98.8 (86.60, 113.4)/cm, p = <0.0001, E) Fasting TG, 136 (98.50,157.8) vs 86.5 (62.00, 107.0)/, mg/dL p = 0.0005, F) BMI, 37.1 (32.85, 41.80) vs 29.03 (23.89, 33.98)/ kg/m², p = <0.0001, G) HOMA2 beta cell function_B, 212.8 (181.0, 253.4) vs 109.2 (90.90, 133.0)/%B, p = <0.0001, H) HOMA2 insulin sensitivity_S, 34.1 (23.38, 42.55) vs 99.6/%S, p = <0.0001, I) HemoglobinA1c, 5.2 (5.000, 5.675) vs 5.6 (4.900, 5.500)/%, p=0.155, J) Fasting LDL, 99 (86.00, 122.3) vs 105.5 (88.00, 123.0)/ mg/dL p = 0.545, K) Fasting Total Cholesterol, 165 (144.5, 191.3) vs 175.5 (152.0, 201.5)/ mg/dL, p = 0.290, IR, Insulin resistance, HDL; High-Density Lipoprotein, TG; Triglycerides, TNF-α; Tumour Necrosis Factor-alpha, HOMAIR; Homeostatic Model Assessment of Insulin Resistance, HOMA2; Homeostatic Model Assessment (version 2), IL5; Interleukin-5, IL6; Interleukin-6, IL10; Interleukin-10, IL7; Interleukin-7, C52; methylbutyryl, C5; Complement Component 5, C2uM; Acetylcarnitine, C2C3C5ratio; Acylcarnitine Ratio, C3uM; Propionylcarnitine.
(TIF)

**S2 Fig. Characteristics of insulin resistance by inflammatory markers between people with and without IR.** This figure shows the median (interquartile range, IQR) between people with and without IR, respectively: A) MIPalpha1, 180 (141.8, 237.3) vs 155 (114.3, 184.0)/ pgml, p = 0.039, B) CRP, 6275935 (2948899, 10699087) vs 3213787(1038752, 8591048)/ pgml p = 0.0204, C) TNFα receptor 1, 11913 (10759, 15334) vs 11165 (9394, 12710)/ pgml, p = 0.012, D) Highly sensitive CRP 4.800 (2.325, 7.700) vs 2.400 (0.8000,6.400)/ pgml, p = 0.018, E) TNFα receptor 2, 17832 (16582, 19215)

vs 16664 (15399 17750)/, pgml p = 0.0016, F) IL6, 5.370 (3.020, 7.423) vs 2.900 (1.985, 4.628)/ pgml, p = < 0.0001, G) IL4, 0.024 (0.00525, 0.05475) vs 0.033 (0.000, 0.08175)/ pgml, p = 0.675, H) IL13, 2.050 (0.7750, 4.175) vs 1.450 (0.050, 2.950)/ pgml, p = 0.185, I) MCP1, 521.5 (407.3, 582.8) vs 451 (387.5, 542.0)/ pgml, p = 0.078, J) Interferon gamma, 18.2 (12.68, 28.13) vs 16.8 (12.95, 26.33)/ pgml, p = 0.712, K) MIP1beta, 298.5 (202.8, 336.0) vs 262 (211.5, 348.3)/ pgml, p = 0.445, L) IL1beta, 0.310 (0.1145, 0.445) vs 0.23 (0.125, 0.391)/ pgml, p = 0.596, M) ICAM1, 548417 (443116, 644533) vs 502401 (416088, 602580)/ pgml, p = 0.253, N) VCAM1, 579836 (458192, 693304) vs 552698 (427486, 616831)/ pgml, p = 0.301, O) Serum Amyloid A, 2573068 (1554338, 5714894) vs 1765767 (895165, 5971649)/ pgml, p = 0.153, P) IL10, 1.02 (0.745, 1.245) vs 0.94 (0.693, 1.518)/ pgml, p = 0.982, Q) IL8, 19.7 (14.3, 30.53) vs 17.65 (15.33, 26.35)/ pgml, p = 0.776, R) IL17, 63.6 (51.05, 111.4) vs 55 (27.15, 80.25)/ pgml, p = 0.052, S) IL2, 0.42 (0.1058, 0.8665) vs 0.59 (0.172, 0.787)/ pgml, p = 0.464, T) TNFalpha, 8.9 vs 8.2 (7.670, 10.880)/ pgml, p = 0.0565. IR, Insulin resistance, TNF-α; Tumour Necrosis Factor-alpha, IL5; Interleukin-5, IL6; Interleukin-6, IL10; Interleukin-10, IL7; Interleukin-7, C52; methylbutyryl, C5; Complement Component 5.
(TIF)

**S3 Fig. Characteristics of insulin resistance by amino acid markers between people with and without IR.** This figure shows the median (interquartile range, IQR) between people with and without IR, respectively: A) Tryptophan, 33.77 (28.17, 39.94) vs 30.13 (25.07, 35.51)/ uM, p = 0.034, B) Leucine, 106.3 (93.99, 121.5) vs 95.96 (82.96, 112.9)/ uM p = 0.013, C) Alanine, 302.4 (256.0356.5) vs 261 (216.0, 303.4)/ uM, p = 0.004, D) Glutamic acid 41.9 (33.19, 74.80) vs 35.4 (27.74, 43.47)/ uM, p = 0.001, E) Isoleucine, 59 (52.21, 71.31) vs (44.23, 62.13) 51.7/, uM p = 0.0041, F) Asparagine, 30.1 (26.92, 36.23) vs 34 (31.30, 38.70)/ uM, p = 0.0051, G) Tyrosine, 53.8 (40.17, 64.59) vs 42.9 (33.74, 51.18)/ uM, p = 0.0006, H) Glycine, 186.0 (156.9, 235.5) vs 231.3 (196.5, 268.0)/ uM, p = 0.0008, I) Arginine, 27.75 (22.14, 35.00) vs 29.61 (24.35, 35.49)/ uM, p = 0.393, J) Citrulline, 30.1 (23.30, 32.62) vs 34.1 (25.47, 37.61)/ uM, p = 0.0051, K) Glutamine, 336.9 (241.0, 386.9) vs 361.2 (307.4, 431.4)/ uM, p = 0.0719, L) Histidine, 64.9 (58.26, 72.50) vs 69.9 (61.76, 74.53)/ uM, p = 0.0995, M) Ornithine, 43.3 (31.58, 52.53) vs 41.2 (34.22, 48.80)/ uM, p = 0.6551, N) Aspartic acid, 4.5 (3.208, 5.466) vs 3.6 (2.615, 5.215)/ uM, p = 0.133, O) Phenylalanine, 42.5 (38.10, 47.82) vs 40.8 (34.80, 45.24)/ uM, p = 0.0766, P) Proline, 137.3 (113.8, 175.4) vs 126.4 (98.72, 152.1)/ uM, p = 0.0719, Q) @3Methylhistidine, 7.5 (4.139, 15.43) vs 9.5 (3.428, 16.97)/ uM, p = 0.6397, R) Serine, 69.9 (57.96, 82.86) vs 77.6 (61.77, 85.89)/ uM, p = 0.275, S) Threonine, 106.9 (83.33, 130.3) vs 104.5 (84.21, 129.5)/ uM, p = 0.8498, T) @1Methylhistidine, 2.73 (2.143, 3.528) vs 2.92 (2.375, 3.412)/ uM, p = 0.561, U) Valine, 186.8 (161.5, 218.0) vs 174.2 (147.2, 209.4)/ uM, p = 0.0602. IR, Insulin resistance.
(TIF)

**S4 Fig. Characteristics of insulin resistance by organic acids and acylcarnitine markers between people with and without IR.** This figure shows the median (interquartile range, IQR) between people with and without IR, respectively: A) C4Butyryl, 0.088 (0.0718, 0.1161) vs 0.069 (0.0461, 0.083)/ uM, p = 0.0019, B) C4Isobutyryl, 0.0498 (0.036, 0.083) vs 0.068 (0.046, 0.085)/ uM p = 0.2687, C) C4DCSuccinyl, 0.007995 (0.006, 0.012) vs 0.00768 (0.006, 0.0099)/ uM, p = 0.472, D) C4OHButyryl 0.0055 (0.005, 0.012) vs 0.005 (0.005, 0.017)/ uM, p = 0.419, E) C5Isovaleryl, 0.0736 (0.054, 0.0899) vs 0.0553 (0.042, 0.0796)/, uM p = 0.0118, F) C52Methylbutyryl, 0.036 (0.0252, 0.0421) vs 0.028 (0.023, 0.0371)/ uM, p = 0.0305, G) C5Valeryl, 0.0030 (0.00305, 0.0043) vs 0.0025 (0.0025, 0.0035)/ uM, p = 0.028, H) 2Hydroxybutyric acid, 31.9 (28.14, 45.74) vs 34.0 (24.34, 48.06)/ uM, p = 0.997, I) Lactate, 1112 (867.5, 1317) vs 919.5 (739.8, 1076)/ uM, p = 0.0028, J) Pyruvate, 106.8 (79.55, 152.0) vs 90.5 (72.03, 110.0)/ uM, p = 0.0405, K) 3Hydroxybutyricacid, 41.16 (31.33, 70.85) vs 53.99 (31.00, 131.5)/ uM, p = 0.301, L) C2, 5.64 (4.917, 6.665) vs 6.79 (4.588, 9.435)/ uM, p = 0.097, M) C3, 0.3113 (0.2246, 0.4025) vs 0.2773 (0.1983, 0.3499)/ uM, p = 0.0518, N) C2C3C5ratio, 13.83 (10.18, 19.11) vs 17.86 (12.51, 29.56)/ uM, p = 0.0045, IR, Insulin resistance, C52; methylbutyryl, C2uM; Acetylcarnitine, C2C3C5ratio; Acylcarnitine Ratio, C3uM; Propionylcarnitine.
(TIF)

**S5 Fig. Forest plot for multivariable regression associations between selected metabolic and inflammatory markers and IR stratified by phenotype and adjusted for key covariates.** Panel A presents a forest plot showing the overall associations of key metabolites, including isovaleryl (C5), glutamic acid, glycine, tryptophan, and tyrosine, with IR, adjusted for age, sex, and BMI. Panel B highlights markers significantly associated with IR in individuals with obesity, including IL-6, fasting HDL, glutamic acid, glycine, IL-5, TNF-α, IL-17, leptin, and MCP-1, adjusted for age and sex. Panel C depicts inflammatory markers—TNF-α1, IL-17, VCAM-1, ICAM-1, leptin, and MCP-1, associated with IR in the overall sample, adjusted for age, sex, and BMI. Panel D shows the corresponding associations in non-obese individuals, where C3uM, leptin, averaged hip circumference, tyrosine, tryptophan, and glycine were significantly linked to IR, adjusted for age and sex.
(TIF)

## Author contributions

**Conceptualization:** Situmbeko Liweleya, Sepiso K. Masenga.

**Data curation:** John R. Koethe.

**Formal analysis:** Situmbeko Liweleya, Sepiso K. Masenga.

**Funding acquisition:** John R. Koethe, Annet Kirabo, Sepiso K. Masenga.

**Investigation:** Sepiso K. Masenga.

**Methodology:** Sepiso K. Masenga.

**Resources:** Sepiso K. Masenga.

**Software:** Sepiso K. Masenga.

**Supervision:** Annet Kirabo, Sepiso K. Masenga.

**Validation:** Situmbeko Liweleya, Brian Halubanza, Lweendo Muchaili, Bislom C. Mweene, Lukundo Siame, Propheria C. Lwiindi, Benson M. Hamooya, Joreen P. Povia, Freeman M. Chabala, Musalula Sinkala, John R. Koethe, Annet Kirabo, Sepiso K. Masenga.

**Visualization:** Situmbeko Liweleya, Brian Halubanza, Lweendo Muchaili, Bislom C. Mweene, Lukundo Siame, Propheria C. Lwiindi, Benson M. Hamooya, Joreen P. Povia, Freeman M. Chabala, Musalula Sinkala, John R. Koethe, Annet Kirabo, Sepiso K. Masenga.

**Writing – original draft:** Situmbeko Liweleya, Brian Halubanza, Lweendo Muchaili, Bislom C. Mweene, Lukundo Siame, Propheria C. Lwiindi, Benson M. Hamooya, Joreen P. Povia, Freeman M. Chabala, Musalula Sinkala, John R. Koethe, Annet Kirabo, Sepiso K. Masenga.

**Writing – review & editing:** Situmbeko Liweleya, Brian Halubanza, Lweendo Muchaili, Bislom C. Mweene, Lukundo Siame, Propheria C. Lwiindi, Benson M. Hamooya, Joreen P. Povia, Freeman M. Chabala, Musalula Sinkala, John R. Koethe, Annet Kirabo, Sepiso K. Masenga.

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
