## [Decision Letter · Decision Letter 0]

11 Nov 2025

PGPH-D-25-02739

Metabolic and Inflammatory Axes of Insulin Resistance in Normoglycemic Adults from the Observational Cohort Study of Adipose Tissue and Immune Activation study: Adipocytokines, Lipids and Amino Acids Pathways

Dear Dr. Masenga,

Thank you for submitting your manuscript to PLOS Global Public Health. After careful consideration, we feel that it has merit but does not fully meet PLOS Global Public Health’s publication criteria as it currently stands. Therefore, we invite you to submit a revised version of the manuscript that addresses the points raised during the review process.

Please pay close attention to the statistical recommendations of the reviewers, particularly addressing whether all associations are adjusted for confounding (and which ones).

We look forward to receiving your revised manuscript.

Kind regards,

Gerard Bryan Gonzales

Academic Editor

Journal Requirements:

i. Please clarify all sources of financial support for your study. List the grants, grant numbers, and organizations that funded your study, including funding received from your institution. Please note that suppliers of material support, including research materials, should be recognized in the Acknowledgements section rather than in the Financial Disclosure. 

ii. State the initials, alongside each funding source, of each author to receive each grant. For example: "This work was supported by the National Institutes of Health (####### to AM; ###### to CJ) and the National Science Foundation (###### to AM)."

iii. State what role the funders took in the study. If the funders had no role in your study, please state: “The funders had no role in study design, data collection and analysis, decision to publish, or preparation of the manuscript.”

iv. If any authors received a salary from any of your funders, please state which authors and which funders.

2. We have noticed that you have uploaded Supporting Information files, but you have not included a list of legends. Please add a full list of legends for your Supporting Information files after the references list.

3. In the online submission form, you indicated, “The data that support the findings of this study are available on request from the corresponding author. The data are not publicly available due to privacy or ethical restrictions.”

3. Uploaded as supplementary information.

Reviewers' comments:

Reviewer's Responses to Questions

**Comments to the Author**

1. Does this manuscript meet PLOS Global Public Health’s publication criteria?

Reviewer #1: Partly

Reviewer #2: Yes

2. Has the statistical analysis been performed appropriately and rigorously?

Reviewer #1: No

Reviewer #2: Yes

3. Have the authors made all data underlying the findings in their manuscript fully available (please refer to the Data Availability Statement at the start of the manuscript PDF file)?

Reviewer #1: Yes

Reviewer #2: Yes

4. Is the manuscript presented in an intelligible fashion and written in standard English?

Reviewer #1: Yes

Reviewer #2: Yes

Reviewer #1: Liweleya et al performed a cross-sectional study of insulin resistance (IR) in normoglycaemic adults. They focused on markers of metabolic, inflammatory, and adipokine pathways to demonstrate which were independent determinants of IR. I have a few comments that could improve the manuscript.

Title: The paper could benefit from a simpler title.

Line 96: The authors stated that HOMA-IR is well validated and offer Ref# 4 (where they studied Chinese participants) as proof. However, there is a body of literature which disputes this. Please see BMC Res Notes. 2014;7:98. doi: 10.1186/1756-0500-7-98 and Diabetes Care. 2013, 36 (4): 845-853. 10.2337/dc12-0840. They need to discuss this as a limitation.

The paragraph starting on line 124 seems better placed before line 114, and lines 132-135 seem redundant.

Line 172: “Adipokines (leptin, adiponectin) and inflammatory markers were measured.” Please give the assays coefficients of variation (CVs) or sensitivities. This would also apply to the organic acids.

Line 175: HOMA indices were calculated by the simple equations. However, the Oxford group has generally recommended using their online calculator to derive model-derived estimates rather than linear approximations. But then in line 202 the authors said they used the calculator. So, please clarify what was actually done. Finally, Lines 182-5 are redundant given they are describing what was said before in Line 175.

Line 228: The multivariable models are not described well enough for easy understanding on how the models were derived/created and the results generated. Could this be reworded?

Line 240: Please define how you made a continuous variable, HOMA-IR, into a dichotomous variable. Were you using the upper quintile of the distribution or some other cutoff? Please justify why the cutoffs were chosen.

Line 244: “Significant difference was also observed based on HIV status…” But these participants were all on HAART medications with viral suppression. So HIV status may be a marker of drug exposure and the drugs were exerting metabolic effects.

Table 2 showed be labelled with the variables used in the model, i.e. age, sex, BMI. Also, would not waist be a better variable to adjust for fat mass, since it is more specific for visceral and ectopic fat than BMI?

Figures: There is an overly large amount figures. Some of these could be summarized in the text or as supplementary figures.

Line 507: Please discuss that reverse causality could also be an explanation of the associations as IR would directly lead to increased gluconeogenesis supported by protein turnover.

Reviewer #2: General review comments to the authours

The manuscript titled “Metabolic and Inflammatory Axes of Insulin Resistance in Normoglycemic Adults from the Observational Cohort Study of Adipose Tissue and Immune Activation: Adipocytokines, Lipids and Amino Acids Pathways” (PGPH-D-25-02739) presents a rigorous, well-organized, and ethically conducted investigation into the multifactorial mechanisms of insulin resistance (IR) in normoglycemic adults. The integration of metabolic, inflammatory, and adipokine markers represents a valuable contribution to metabolic and public health research.

1. Scientific and technical merit

The study is well-conceived and addresses a significant knowledge gap on early metabolic dysfunction before the onset of diabetes. The manuscript demonstrates methodological soundness and aligns with PLOS Global Public Health’s focus on translational and population-relevant research.

L43–L83 (Abstract): The abstract clearly presents the study background, objectives, methods, and conclusions. However, the prevalence of IR (L62–L63) is repeated and can be stated once for conciseness. The concluding sentence (L75–L79) effectively highlights the emergence of two IR phenotypes and should remain as the key takeaway.

L87–L135 (Introduction): The rationale and context are strong, but the first two paragraphs (L87–L99) can be condensed to avoid redundancy regarding the roles of adipokines and cytokines. The study hypothesis (L114–L121) is clear and logically follows the background provided.

L124–L135: The mention of methodological variability in insulin resistance assessment is appropriate; however, citing comparative studies on HOMA2-IR validation would further strengthen this point.

Overall, the study presents a clear rationale and scientifically valid objectives that are appropriately addressed by the chosen methods.

2. Statistical analysis

The statistical methods are appropriate and executed rigorously, supporting the validity of the conclusions drawn.

L211–L231 (Data Analysis Plan): The statistical workflow, including the use of Shapiro–Wilk test, Chi-square, Fisher’s exact, Pearson’s, and Spearman’s correlations, is well justified.

The multivariable linear regression model (L225–L230) is correctly structured, but it would be helpful to state whether covariates were entered simultaneously or through stepwise selection.

L365–L440 (Regression Analysis): Table 2 effectively presents β-coefficients and confidence intervals. Ensure consistency between univariate and multivariable models and verify that all adjustments (age, sex, BMI) were retained across analyses. Reporting adjusted R² values for transparency would strengthen interpretation.

L443–L459 (Figure 9): The forest plot is an effective visual summary. Confirm that all variables plotted correspond to those in Table 2, and that abbreviations (e.g., C3uM, IL-17, MCP-1) are fully defined in the legend.

The statistical evidence adequately supports the identification of two mechanistically distinct IR phenotypes.

3. Ethical and data considerations

L137–L145 (Ethics Statement): Ethical approval (IRB# 121175) and informed consent are appropriately documented. The authors followed the principles of the Declaration of Helsinki and ensured participant confidentiality, fulfilling PLOS ethical standards.

Data Availability Statement (page 3): The authors indicate that data are available upon request due to privacy and ethical restrictions. While acceptable for sensitive datasets, this approach is partly compliant with PLOS’s open-data policy. The authors should deposit a de-identified dataset or summary-level data in a public repository (e.g., Zenodo or Dryad) and include the repository DOI in the final version.

4. Clarity and presentation

The manuscript is written in clear, standard academic English, with strong organization and logical flow between sections.

L239–L247 (Results – Participant Characteristics): The data are well described but could be presented more succinctly; the same statistics appear in both text and Table 1. Retain the values in the table and summarize patterns in the text.

L281–L285 (Inflammatory Profile): The findings are clearly reported; ensure consistent presentation of units (pg/mL, µM) throughout.

L311–L316 (Amino Acids): The narrative is clear and well-supported by Figure 3.

L463–L486 (Discussion – Main Findings): The summary of key findings is concise and well aligned with the data.

L501–L523 (Discussion – Mechanistic Pathways): This section is detailed and scientifically sound but could be reduced slightly to improve readability.

L529–L553 (Conclusion): The conclusion appropriately synthesizes the study’s implications. Consider linking the findings to practical applications in early diabetes screening or metabolic risk assessment.

Minor editorial points:

Maintain consistency in abbreviation definitions (IL-6, TNF-α, HOMA2-IR).

Check uniformity of units (mg/dL vs. mmol/L) and ensure SI compliance.

Correct minor typographical issues (spacing, punctuation in tables).

5. Contribution and relevance

The study advances understanding of early insulin resistance and distinguishes between metabolic and inflammatory-lipotoxic pathways. The identification of glycine as inversely associated with IR and propionylcarnitine (C3uM) as a metabolic biomarker offers translational potential. These findings have important public health implications, suggesting early biomarkers for metabolic risk among normoglycemic individuals, especially in populations with high HIV prevalence.

6. Summary recommendations

The manuscript is technically sound, ethically compliant, and clearly written. The conclusions are well supported by the data. With minor editorial refinements and compliance with open-data requirements, this study will make a strong contribution to the literature on metabolic and inflammatory determinants of insulin resistance.

Recommendation: Accept with Minor Revisions

Required Minor Revisions:

• Condense repetitive numerical reporting in the Results section (L239–L247, L281–L285).

• Add details on the variable entry approach in regression modelling (L225–L230).

• Standardize units and abbreviations throughout.

• Deposit de-identified data or summary tables in a public repository and update the Data Availability Statement.

• Reduce background repetition in the Introduction (L87–L99) and Discussion (L501–L523).

**Do you want your identity to be public for this peer review?** For information about this choice, including consent withdrawal, please see our Privacy Policy

Reviewer #1: No

Reviewer #2: **Yes: ** Themba Sigudu

---

## [Editor Report · Decision Letter 1]

1 Dec 2025

PGPH-D-25-02739R1

Metabolic and Inflammatory Axes of Insulin Resistance in Normoglycemic Adults

Dear Dr. Masenga,

Thank you for submitting your manuscript to PLOS Global Public Health. After careful consideration, we feel that it has merit but does not fully meet PLOS Global Public Health’s publication criteria as it currently stands. Therefore, we invite you to submit a revised version of the manuscript that addresses the points raised during the review process.

I appreciate the effort in answering the reviewers' comments. However, I still believe that one point needs to be clarified. This population is very specific and not generalizable to normoglycemic populations: 70% of participants have HIV. This should be emphasized more and declared, from the title already, that the study is conducted in a population of high HIV burden. The discussion section should therefore not directly relate its observations to other studies without stressing the potential interaction of HIV in this context. Minor revision is requested to avoid readers from misunderstanding the results to be generally applicable to normoglycemic populations.

We look forward to receiving your revised manuscript.

Kind regards,

Gerard Bryan Gonzales

Academic Editor
---

## [Editor Report · Decision Letter 2]

9 Dec 2025

Metabolic and Inflammatory Axes of Insulin Resistance in Normoglycemic Adults with High HIV Burden

PGPH-D-25-02739R2

Dear Prof. Masenga,

We are pleased to inform you that your manuscript 'Metabolic and Inflammatory Axes of Insulin Resistance in Normoglycemic Adults with High HIV Burden' has been provisionally accepted for publication in PLOS Global Public Health.

Best regards,

Gerard Bryan Gonzales

Academic Editor